# Deep Learning-Based Artificial Intelligence to Investigate Targeted Nanoparticles’ Uptake in TNBC Cells

**DOI:** 10.3390/ijms232416070

**Published:** 2022-12-16

**Authors:** Rafia Ali, Mehala Balamurali, Pegah Varamini

**Affiliations:** 1School of Pharmacy, Faculty of Medicine and Health, University of Sydney, Sydney, NSW 2006, Australia; 2Australian Centre for Field Robotics, The University of Sydney, Sydney, NSW 2006, Australia; 3The University of Sydney Nano Institute, The University of Sydney, Sydney, NSW 2006, Australia

**Keywords:** artificial intelligence, machine learning, image classification, nanoparticles, triple negative breast cancer, drug cellular uptake

## Abstract

Triple negative breast cancer (TNBC) is the most aggressive subtype of breast cancer in women. It has the poorest prognosis along with limited therapeutic options. Smart nano-based carriers are emerging as promising approaches in treating TNBC due to their favourable characteristics such as specifically delivering different cargos to cancer cells. However, nanoparticles’ tumour cell uptake, and subsequent drug release, are essential factors considered during the drug development process. Contemporary qualitative analyses based on imaging are cumbersome and prone to human biases. Deep learning-based algorithms have been well-established in various healthcare settings with promising scope in drug discovery and development. In this study, the performance of five different convolutional neural network models was evaluated. In this research, we investigated two sequential models from scratch and three pre-trained models, VGG16, ResNet50, and Inception V3. These models were trained using confocal images of nanoparticle-treated cells loaded with a fluorescent anticancer agent. Comparative and cross-validation analyses were further conducted across all models to obtain more meaningful results. Our models showed high accuracy in predicting either high or low drug uptake and release into TNBC cells, indicating great translational potential into practice to aid in determining cellular uptake at the early stages of drug development in any area of research.

## 1. Introduction

Breast cancer is the most prevalent malignancy and the leading cause of death in women worldwide. Among the different subtypes of breast cancers, triple negative breast cancer (TNBC) accounts for 15–20% and constitutes 40% of deaths within the first five years of diagnosis [1,2]. The complex nature of the subtype relates to its unique immunohistochemical characteristic revealing a lack of estrogen receptors (ER), progesterone receptors (PR), and human epidermal growth factor 2 (HER2) receptors on the tumour’s membrane [2]. As a result, TNBC is not sensitive to hormone-based therapy and HER2 directed treatment, leaving chemotherapy, radiotherapy, and surgery as the main current treatment modalities [3]. The mainstay of pharmacotherapy in neoadjuvant, adjuvant, and metastatic settings is non-specific chemotherapy, which is associated with substantial adverse events due to the off-target effects [2]. This drastically impacts patients’ quality of life. Despite TNBC being one of the greatest responders to chemotherapy initially, there is inherent and subsequently acquired resistance to treatment [2]. All factors lead to poor patient prognosis, high risk of metastasis, and relapse [4]. Therefore, developing other therapeutic approaches in treating TNBC patients is critical.

Nano-based carriers are emerging drug delivery systems that are extensively being explored in cancer treatment. The potential integration of nanotechnology in the treatment of TNBC is driven by their auspicious characteristics and preferential accumulation into tumour tissues. Selectivity is achieved through both passive and active means [5]. Passive targeting relates to the enhanced penetration and retention (EPR) effect that relies on the leaky vasculature of tumour blood vessels [6]. Whilst engineered moieties on the surface of the nanoparticles (NPs) contribute to active targeting [5]. Suitable surface functionalising candidates include amino acids, vitamins, antibodies, and peptides [5]. Precision therapy occurs through increased drug uptake in the cancer cells, in conjunction with delivering consistent quantities of medication in a controlled manner [7]. Furthermore, the limited biodistribution to non-cancerous tissue improves the side effect profile, further establishing their benefit. This study used images showing the uptake of polymeric NPs loaded with a fluorescent anticancer payload (green). These NPs were functionalised with a naturally occurring targeting peptide moiety. The targeting moiety or the naturally expressed peptide within the cells were detected through immunofluorescence using a fluorescently labelled secondary antibody (Rhodamine, red) against the primary antibody binding to the peptide (Figure 1).

During the nanopharmaceutical development process in oncology, analysis of the drug candidates’ cellular uptake is an integral component. Antitumour therapeutics are active inside the cell, but occasionally the cellular membrane poses as a barrier to their entry due to its relatively impermeable structure [8]. The major route of internal cell access for NPs is through endocytosis [5]. Functionalisation with targeting molecules and optimum size can facilitate a higher uptake in specific cells, e.g., tumour cells, through targeting. A common qualitative technique employed to determine cellular uptake and localisation of the NPs within different structures of the cell, is confocal laser scanning microscopy (CLSM) [9]. NPs and cell components are labelled with different fluorophores, each colour assigned to a different factor to allow for visual differentiation [8]. However, as the interpretation of images is subjective to the expertise level of the analyser, it can lead to substantial human error and bias. An additional limitation of the method is that the evaluation of uptake through image analysis is time-consuming. To mitigate these issues, we proposed that automating the process through artificial intelligence (AI) will provide high accuracy results as either high or low cellular drug uptake in a timely fashion.

Motivated by the capability of handling large data volumes, AI has been increasingly used in data digitalisation in the pharmaceutical sciences to solve complex clinical issues. AI encompasses various domains, such as knowledge representation, reasoning, solution search, as well as a key paradigm called machine learning (ML) [10]. Deep learning (DL) is a branch of machine learning that simulates the behaviours of the human brain [11]. It imitates the natural neural network; the way human neurons and their connections retrieve and process diverse forms of inputs to form a conclusion [12]. DL algorithms consist of multiple layers that have interconnected nodes; each node builds on previous nodes, refining information to make accurate predictions [13]. The visible layers of the deep neural network are known as input and output layers (Figure 2). Input is what the model trains on, and the output is a prediction made by the model following training. DL algorithms have been successfully applied in medical imaging and disease detection, identification, and diagnosis. For example, in breast cancer, several studies have reported the utilisation, or the potential application, of automation in improving early detection and diagnosis with subsequent classification of cancer subtypes [14,15,16,17,18,19,20,21,22,23,24,25]. DL models have also been studied in creating treatment plans for locally advanced breast cancers and predicting survival rates [26,27]. However, the integration of AI in drug development is an area that is still not well established. Our study aimed to develop a method using DL models that could rationally allow the selection of the best drug candidate based on their cellular uptake and via qualitative analysis, a critical step during the drug development process. Herein, we used TNBC as the disease model and we investigated the uptake of nanoparticles with a fluorescent (green) payload.

## 2. Results

Convolutional neural networks (CNN’s) are DL algorithms that have become state-of-the-art in image classification projects with superior results in the clinical setting [28]. In this study, five different CNN models were compared: two sequential models from scratch and three pre-trained models, VGG16, ResNet50, and Inception V3. To evaluate the performance of our models, various approaches were used. Model performance and cross-validation accuracy were conducted as initial accuracy measures. A 5-fold cross validation report was then generated for each model and for both the high and low classes. Confusion metrics and receiver operating characteristic curve (ROC) were also produced to observe the accuracy of the predictions made by the models. Lastly, a comparative analysis between the best performing model and the conventional method to determine drug cellular uptake was conducted to further validate efficacy, as extended on in the following section.

### 2.1. Model Performance

The accuracy results of the CNN models are presented below (Table 1). All models had relatively high accuracy, but the pre-trained Inception V3 model had a pronounced performance for the binary classification, with an accuracy rate of 99.35%. The result implies that the model accurately predicted 99.35% of the test input as being contained in either a high or low class. Both models from scratch performed comparatively well and had similar accuracy results. Model training and validation accuracy followed a desirable increase trend per epochs, as shown in Appendix A.

The cross-validation classification report based on class was also determined, as summarised in Table 2. Overall, models showed a higher precision for images with low drug uptake, in the low classes, with Inception V3, again, generally outperforming the other models in various measures. The model had the highest precision for the high class of 1 and the low class of 0.986. It also had the most heightened sensitivity for the high category of 1. However, for the low-class VGG16 had better results for both sensitivity and specificity, 0.992 and 0.990, respectively. Lastly, the f-1 score was highest with Inception V3 for the high class as 0.994 and the low class as 0.996. In general, except for precision, VGG16 performed poorly on high-class images and ResNet50 on images from the low class.

An additional approach to evaluate accuracy was by producing a ROC for the last fold of each models training (Figure 3). The AUC was also measured for each one. Overall, all models performed exceedingly well, with all having an AUC of 99%. Meaning the models can predict the output with 99% accuracy, whereas the confusion metric, as shown in Figure 4, displays the predictions per class. Inception V3 generated an output of 163 of the high intensity signals as high and 0 as low. Whilst for the low class, 203 images were correctly identified as having low drug uptake, whereas two images of the low signals were predicted as high cellular uptake.

### 2.2. Comparison Using Manual Intensity Evaluation and Predictions from AI

To assess the validity of the DL models’ performance, a comparison of the predictions was made against the conventional method of determining drug cellular uptake using confocal imaging. A confocal image not previously seen by the algorithm was used. The average signal intensity of manual measurement using five sample areas with the program ImageJ was determined to be 48.5. For reference, two untargeted nanoparticle confocal images intensity were evaluated to be 33.61 and 17.15, which indicated high and low, respectively. Based off these values, the average intensity of the comparison image indicates the nano-based drug carrier as having high cell uptake. Random patches from the same image were used to predict using the Inception V3 model, the best performing model in this study. An average of ten predictions was calculated to be 0.742, with a threshold of 0.5. Above 0.5 means high class and below means low class. The results of our experiment also indicate the drug delivery system in the image as having high anticancer drug cell uptake.

## 3. Discussion

The substantial burden of TNBC has made the discovery of potential nanoparticles and their integration into therapy indispensable. As such, quick determination of cellular drug uptake needs to be conducted more efficiently. Automation has shown to be a promising approach in successfully assisting researchers in various settings. Our study has displayed that AI can be effectively trained to accurately determine the level of drug uptake into cells to assist in the drug development process, particularly in a comparative way. For instance, these models can make an accurate comparison between the uptake of targeted vs. untargeted drugs to inform the efficiency of targeting in a particular delivery system.

This study comprehensively evaluated the effectiveness of five different CCN models, two from scratch plus VGG16, ResNet50, and Inception V3. The pre-trained algorithms were used to see if they would have enhanced accuracy since they have been previously exposed to various images and could transfer those learnt features. However, in previous studies, medical images containing multiple colour channels have been hard to differentiate [29]. We observed variation in the cross-validation results between the three models with varying depths in their layers. Our two models from scratch performed significantly well, considering the simplicity of their layers. The pre-trained model Inception V3 did perform the best overall, with highly accurate predictions in both high and low classes. The overall cross validation performance of VGG16 and ResNet50 are poorer compared to the other models, whilst VGG16 was also associated with significant computational cost. The performance gap between these models could be due to their network architecture. It is likely that a search for better hyperparameters for individual models would yield additional improvement. This study also demonstrated CNN’s ability to differentiate multiple-coloured channels, as has been a limitation in previous studies [29].

A challenge encountered in the study was access to limited data. Several of the original confocal images contained significant background noise, which would not allow an accurate representation and evaluation of the uptake. To ensure the algorithm was trained correctly and to avoid confusion, these images were excluded, and only high-resolution images were used as the input data, which considerably limited our input sample size. Nevertheless, as can be seen in the results, all CNN models performed substantially well on the three channel images. An explanation for the high accuracy of the results could be due to the training and validation data patches having originated from the same 2 original images, despite the image patches being distinctly different from each other (Figure 4). Increasing the variety of the original images will help to train the model more efficiently and enhance the model’s ability to distinguish features, allowing the predictions of drug uptake with any new/unseen confocal image in a more appropriate manner [30].

A new confocal image was patched when the model’s predictions were evaluated and compared to the conventional methodology. Subsequently, the patches that contained irrelevant information were removed. As a result, only images that contained the cell nucleus were used to make the predictions.

Manually removing sections of unimportant details is time-consuming, and there is room for potential bias. A viable approach to this issue is a multi-class classification DL model that can identify for instance the blue signal (cell nucleus), disregard the irrelevant areas, whilst leaving the green signal (payload) as an essential feature for detection. With this approach, pre-processing will not be required before the model can predict unseen images, further removing human bias and making it an efficient method to evaluate cellular drug uptake. Furthermore, other approaches to this hypothesis is to use a model, such as YOLO or other segmentation models (Mask-RCNN, UNet are some), that has localized interested region, i.e., blue region. The trained models can detect the cancer cell area automatically instead of manually determine areas of interest within the cell where the uptake can be detected and quantified automatically.

Nevertheless, our study has established a good foundation for integrating artificial intelligence with drug development in various areas, particularly in oncology. It can further be built on to yield easier drug uptake detection and conduct comparative studies to determine the lead drug candidate(s).

## 4. Materials and Methods

In this study, various DL models were trialled to automate determining the cellular uptake of nanoparticles based on their fluorescent payload. The proposed methodology entailed preparing the input data, training pre-determined neural networks, and making predictions of high or low cell uptake. A comparison between the manual process of determining uptake and the computational output using the best performing model was also propositioned. Image classification using pre-trained and sequential models from scratch (i.e., models layers and parameters that we determined) is extended in the following sections.

### 4.1. Data Pre-Processing

#### 4.1.1. Patch Generation

Confocal images were processed into trainable inputs that a DL algorithm could be taught. The objective of patch generation was to increase the input sample and give flexibility to eliminate areas that may be a cause of confusion or may not contain useful information for the training process. The original size (3487 × 3487) of each visual input was patched into pixels of 224 × 224 with three colour channels of blue, red, and green. The blue represents the TNBC cell nucleus, the red fluorescent signal illustrates the functionalised moiety of the nanoparticles or the naturally occurring peptide within the cells, and the green signifies the fluorescent payload. Hence, the concluding input size for the model was 224 × 224 × 3 (height × width × channel) (Figure 5b).

#### 4.1.2. Image Classification

To facilitate supervised learning, images were classified into two descriptive classes, high and low, based on the intensity of the green fluorescent signal, which was conducted under the supervision of a domain expert. The high intensity of the green signal illustrates high uptake of the anticancer agent into the TNBC cell. In contrast, a low fluorescent signal means the uptake of the drug was minimal. The exclusion criteria to ensure proper training included images that contained no valuable input, e.g., images with no blue signal (TNBC nucleus) indicating the image did not represent the inside of the cell. In addition, images that were a probable source of confusion to the matrix (Figure 5d) were consequently removed, resulting in a final input sample of 225.

#### 4.1.3. Data Augmentation Rotation

An additive approach to satisfy the requirement of large input samples for training neural networks is data augmentation. By making pre-determined modifications to existing data, the model is exposed to a greater diversity of learnable features and thus can make more generalised predictions. The augmentation type used in this study was rotation. Each applicable patched image was rotated at 45°, resulting in 7 new images (Figure 5e). This technique successfully enhanced the sample data, totalling 1824 images. Train and test sets were then generated, 80% and 20% of the augmented data, respectively. The training accuracy, loss, and validation accuracy and loss were measured upon training completion.

### 4.2. Prediction Using Pretrained and Scratch Convolutional Neural Network (CNN) Algorithms

The type of CNN used in this project is known as supervised learning which involves the provision of labels associated with specific images, e.g., high and low [31]. The features and patterns are identified, and the algorithm then generalises the input it receives. The fundamental layers of CNN’s perform different computational transformations to their inputs. CNN’s can contain complex layers but mainly consist of convolution layers, pooling layers, and the fully connected layer [32]. Each layer’s output is the subsequent layer’s input features (Figure 6). This feed-forward learning allows CNN applications to acumen on image data. In this study, we trained with two models from scratch and three pre-trained models, which have been elucidated in the following section.

#### 4.2.1. Convolutional Layer

The first layer within a neural network is known as the convolutional layer. It extracts features of the input data and determines learnable patterns [34]. The learned elements are then iterated to the next layer, which is conducted through what is known as a kernel (Figure 7). The kernel convolves over pixels of the input data according to the filter size that is set [35]. The deeper the layer, the more sophisticated detail the filter can detect. The convolutional layer is followed by an activation layer which is involved in the decision-making process. In our project, we used sigmoid and ReLU.

#### 4.2.2. Pooling Layer

Generally, after the convolutional layer is the pooling layer. Its functional purpose is to reduce the dimensionality of images by reducing the number of pixels formed from the output of the previous convolutional layer. Essentially, it receives the input data from the neighbouring group of neurons and summarises the units through either max pooling or average pooling, thus reducing the computational load (Figure 8). By doing so, it can put forward the most activated pixels, allowing the neural network to observe the essential pixels.

#### 4.2.3. Fully Connected Layer

The fully connected (FC) layer, as the name suggests, connects the neurons of the previous layer to the neurons of the following layer [36]. In summary, in a CNN, the convolutional layer extracts features of the image, the pooling layer reduces pixel dimensions, and the fully connected layer maps those features to the predicted outputs.

### 4.3. Training Regime with Model Specifications

#### 4.3.1. Training Small Convnets from Scratch

Initially, simple models with minimal layers that had appropriate varying parameters were used to see if they would yield high accuracy. The two supervised learning models were developed using the Keras sequential models. The architecture of these models is illustrated below. The second model is the more complex of the two, which consists of a simple stack of three convolution layers with a ReLU activation followed by a max-pooling layer (Figure 9). At the end of this pipeline, two fully connected layers are attached. The model ends with a single unit and a sigmoid activation, optimal for binary classification. The model used the binary cross-entropy loss to train the model. A drop out layer was used to minimise the model overfitting during training. Overfitting occurs when the input data is limited, and the model is not able to generalise the features well. So, the training accuracy is high, but the validation accuracy remains low. Model 1 is different from model 2 by the number of convolution layers; only two convolution layers were used in this model (Figure 10).

#### 4.3.2. Using Pre-Trained Models

Another approach in image classification is fine-tuning pre-trained models, also known as transfer learning. The concept of transfer learning is that instead of starting from scratch, models have been previously trained on various images. Those skills are then transferred over to when the model is training and evaluating new data sets. Moreover, they have been designed with specific architectures and pre-defined weights; weights are a value that represents how the connection between each node in a layer facilitate transfer learning. Models from scratch are associated with high computational cost, acting as the driving factor to use these specific pre-trained models. They also require large amounts of data to effectively learn features and then generalise them to make accurate predictions. However, this is not always possible, as in many cases, the available data is limited, which is an issue we encountered. Therefore, we used pre-trained models in this binary classification study which were Inception-V3, ResNet-50, and VGG16 (Figure 11). These models were previously trained on large sets of data and were later fine-tuned on the confocal images. The same optimization configuration (optimizer = Adam and lr = 0.001) was used for all models. This configuration was tuned for best performance with Inception V3 because of its less computational power and we did not attempt to tune optimization hyperparameters for individual models separately.

### 4.4. Model 5-Fold Cross-Validation

The CNN models were evaluated using a test set containing images of both high and low anticancer drug uptake, which had not been observed during the training stage. Each model was previously trained for ten epochs, and a batch size of sixty-four was decided empirically. The training was conducted using Adam as the optimiser as it worked effectively on our three-channel confocal images. Each model’s performance cross-validation was evaluated based on different metrics such as precision, sensitivity specificity, F-1 scores, and confusion metrics (Table 3). ROC curve and area under the curve (AUC) were presented from the last fold of each cross-validation (Figure 3). The parameters of the confusion metrics were used to determine each metric- True Positive (TP), True Negative (TN), False Positive (FP), and False Negative (FN) [38].

## 5. Conclusions

In this study, we proposed integrating DL-based automation in determining drug uptake for cancer research. As nanoparticles have great potential in TNBC, the efficient uptake of these carriers is an essential measure that determines the drug delivery system’s effectiveness. Hence, the need for computer-aided identification of the manual qualitative analyses for drug uptake has driven this research. Several CNN models have been shown to perform substantially well in appropriately predicting nanoparticle uptake and subsequent drug release into the TNBC cell. Of the pre-trained models, Inception V3 produced a better outcome with an overall cross-validation accuracy of 99.34%, and hence regarded as a potential model that can be used in future studies in assessing elements of the drug discovery process. This will allow quicker evaluation of cellular drug uptake and improves the accuracy of this stage in the drug discovery pipeline, whilst also reducing the manual workload associated with this qualitative methodology. The DL algorithms need to be built on to address multi-class categorisation for practical translation to occur. Nevertheless, this research has paved the way for future studies incorporating DL-based automation in drug development.

## Figures and Tables

**Figure 1 ijms-23-16070-f001:**
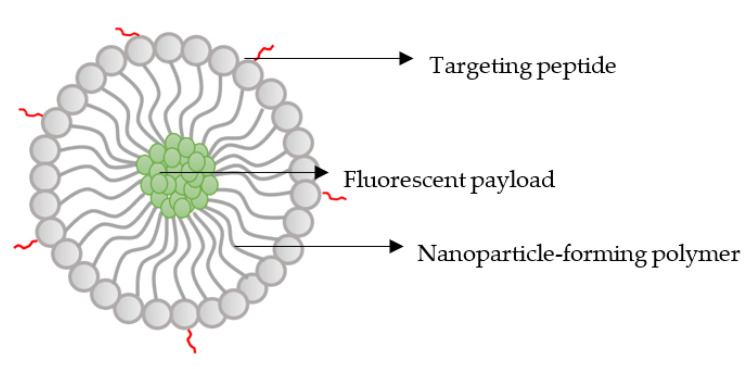
Illustration of a peptide-functionalised NPs loaded with a drug.

**Figure 2 ijms-23-16070-f002:**
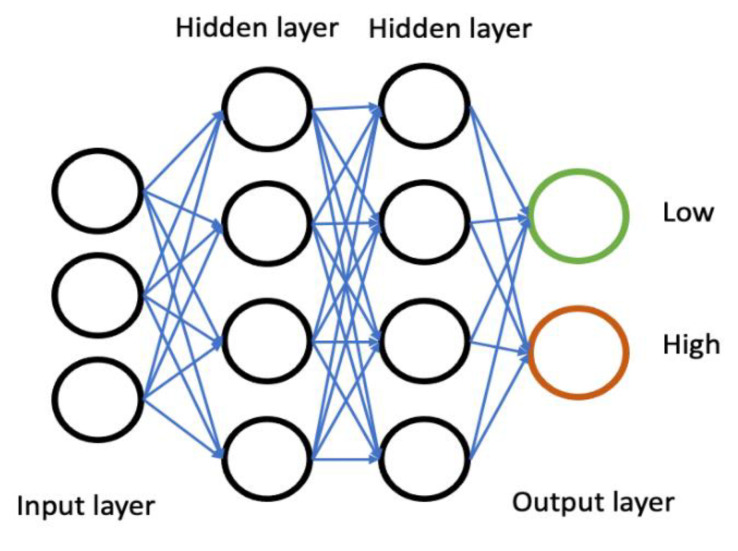
Architecture of deep learning algorithms.

**Figure 3 ijms-23-16070-f003:**
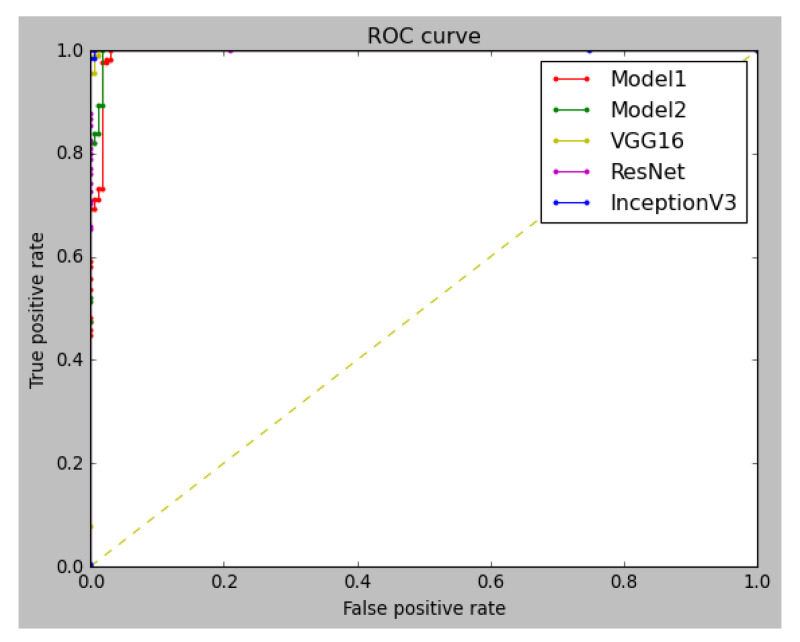
ROC curve of last fold cross-validation sample. (AUC for model 1- 0.994434, Model 2- 0.997247, VGG16-0.999671, ResNet50- 0.994859, and Inception V3- 0.999910).

**Figure 4 ijms-23-16070-f004:**
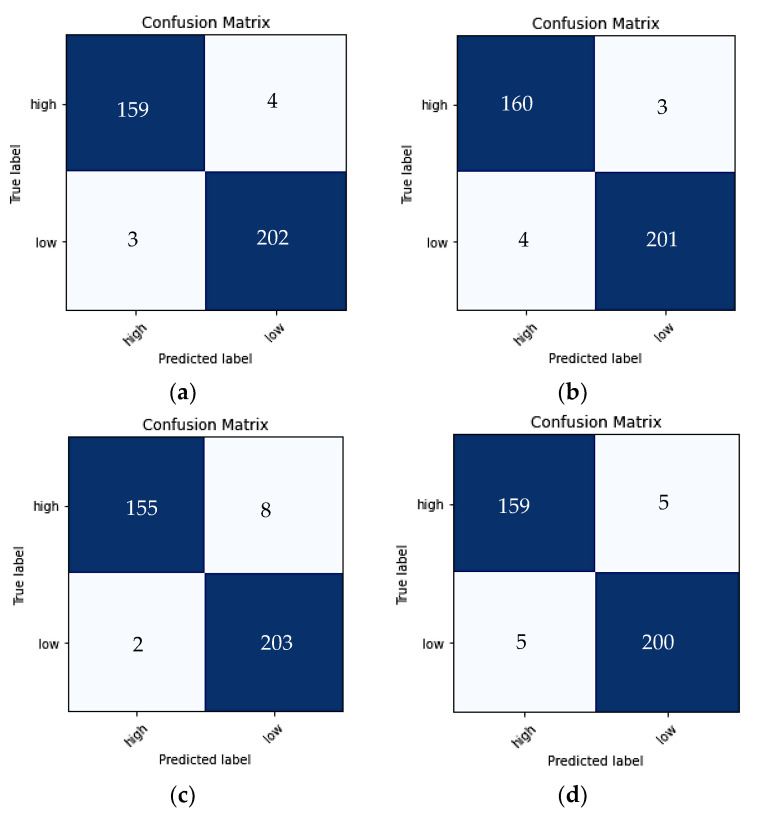
Confusion metrics cross-validation results for (**a**) Model 1 (**b**) Model 2 (**c**) VGG16 (**d**) ResNet50 (**e**) Inception V3.

**Figure 5 ijms-23-16070-f005:**
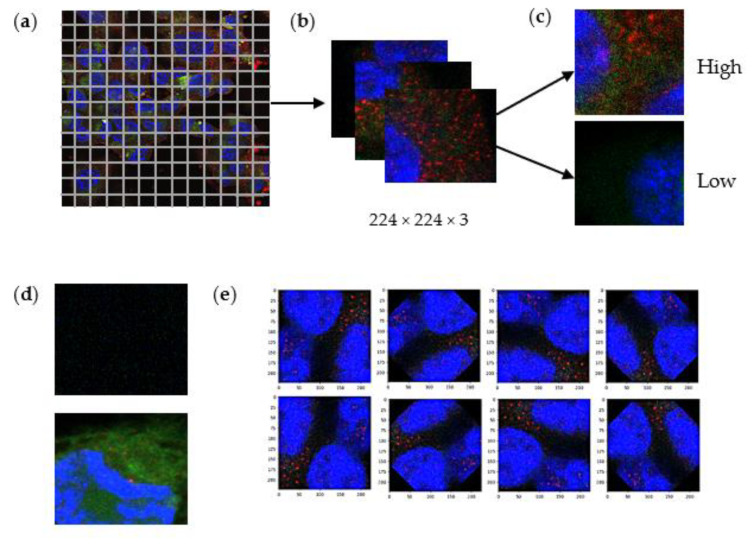
Illustration of data processing technique: (**a**) Patch sizes to 224 by 224 pixels (**b**) Resulting patch images (**c**) Image classification into high or low classes based on the green fluorescent signal (**d**) Excluded images (**e**) Data augmentation of individual patched image.

**Figure 6 ijms-23-16070-f006:**
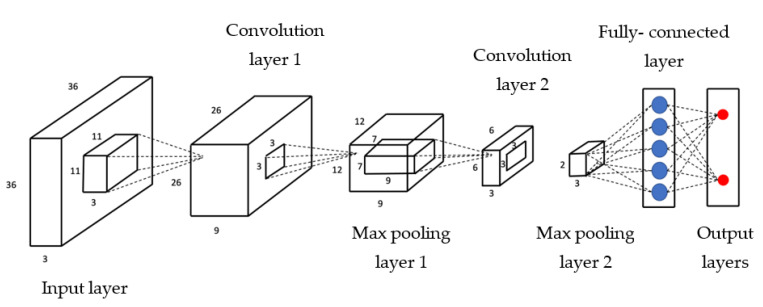
Illustration of a CNN model [33].

**Figure 7 ijms-23-16070-f007:**
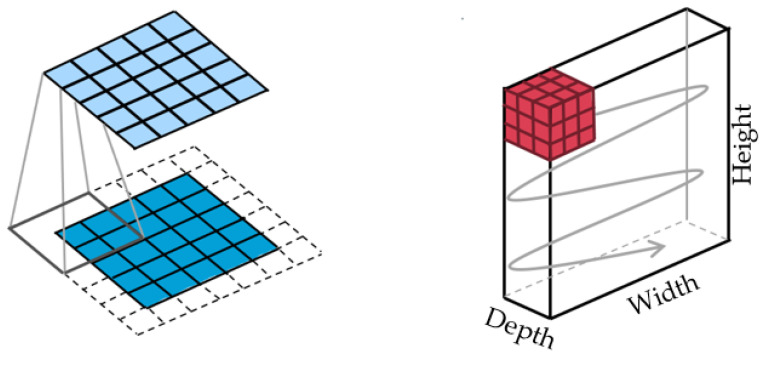
Convolutional layer and the kernel movement pattern [34].

**Figure 8 ijms-23-16070-f008:**
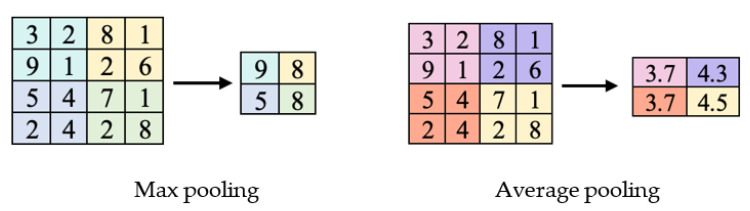
Different pooling types used in CNNs.

**Figure 9 ijms-23-16070-f009:**
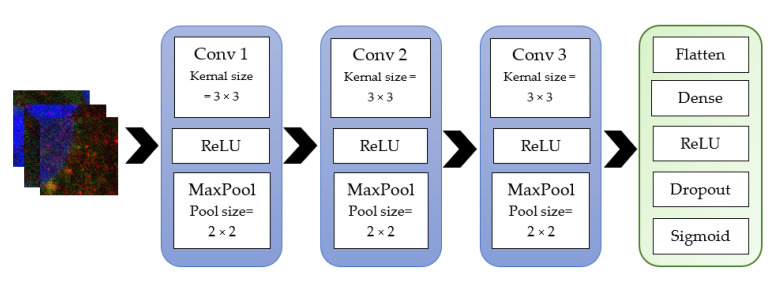
Model 2-sequential model’s layers.

**Figure 10 ijms-23-16070-f010:**
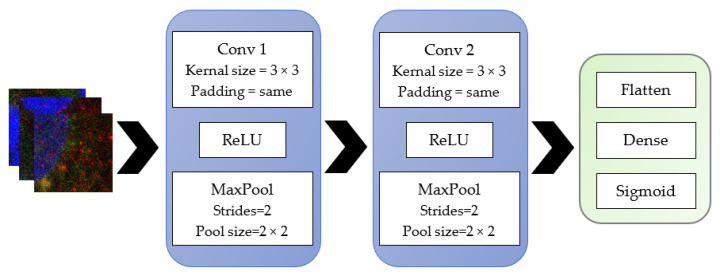
Model 1-sequential model’s layers.

**Figure 11 ijms-23-16070-f011:**
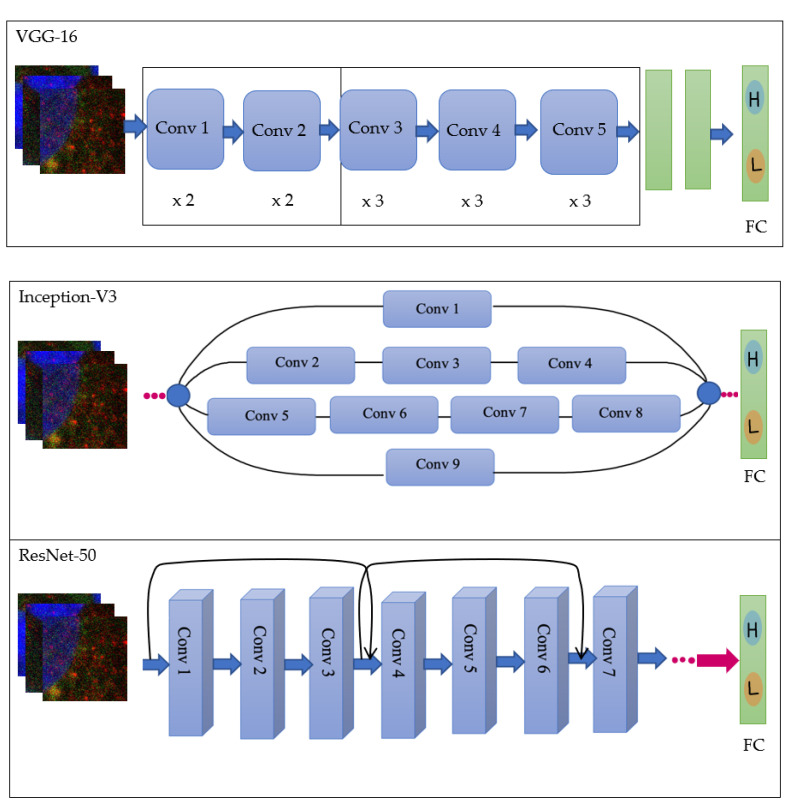
Architecture of the pre-trained models [37].

**Table 1 ijms-23-16070-t001:** Cross-validation accuracy across all models.

	Model 1	Model 2	Model 3 (VGG16)	Model 4 (ResNet50)	Model 5 (Inception)
Fold 1	98.37	97.01	98.64	99.18	98.91
Fold 2	98.91	98.91	98.91	97.83	99.73
Fold 3	99.18	99.46	93.48	95.38	99.73
Fold 4	95.65	96.20	95.92	94.29	99.46
Fold 5	98.37	99.18	99.46	100	98.91
Overall accuracy	98.076	98.152	97.282	97.336	99.348

**Table 2 ijms-23-16070-t002:** 5-fold cross-validation classification report per class.

Model	Class	Precision	Sensitivity	Specificity	F-1 Score
Model 1	High	0.980	0.976	0.975	0.978
	Low	0.980	0.988	0.985	0.984
Model 2	High	0.978	0.982	0.982	0.980
	Low	0.986	0.982	0.980	0.984
VGG16	High	0.992	0.948	0.951	0.966
	Low	0.964	0.992	0.990	0.976
ResNet50	High	0.968	0.970	0.975	0.970
	Low	0.976	0.978	0.976	0.976
Inception-V3	High	0.986	1.00	1.00	0.994
	Low	1.00	0.99	0.990	0.996

**Table 3 ijms-23-16070-t003:** Metric equations.

Metrics	Equation
Precision	TP/FP + TP
Sensitivity	TP/FN + TP
Specificity	TN/FP + TN
Accuracy	TP + TN/ TN + TP + FN + FP
F-1 score	(Precision × recall/ precision + recall) × 2

## Data Availability

Not applicable.

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
