# Peer review of "Deep Learning-Based Artificial Intelligence to Investigate Targeted Nanoparticles’ Uptake in TNBC Cells"

_ijms, 2022, doi:10.3390/ijms232416070_

Round 1

Reviewer 1 Report

Ali et. al. evaluate the performance of five different CNN models in predicting either high or low cell uptake of drug loaded NPs into TNBC cells. The models are trained using confocal images of cells under the treatment of NPs loaded with fluorescence labeled anticancer drug. The authors claim that their models show high accuracy and great translational potential in drug development. However, the models are not newly developed and the application of this study seems narrow. Some concerns are as follows:

1.      The models used in this manuscript are published and well-developed. It seems that the authors didn’t develop any new algorithm but used the published models for comparison, which reduces the novelty of this study.

2.      The authors focus their research on cell uptake of drug loaded NPs in TNBC. The topic is very narrow, which limits the application of this study.

3.      The model exports a result of either high or low uptake. However, there are already commercialized software on confocal microscope providing quantitative fluorescence results based on machine learning. The method seems not powerful.

4.      There are lots of CNN based machine learning methods reported for figure analysis in the field of cancer. The authors should cite more references to summarize what has been accomplished in the field, and then highlight what's the difference and novelty of this study.

5.      Is cross-validation accuracy the single significant parameter to be compared? The differences among the chosen models based on this single parameter are not very significant. More and better comparison strategy should be applied.

6.      What's the dye with red fluorescence? Are NPs also labeled with probe? The authors should provide the information of each dye.

7.      In table 1, what happened to ResNet model in Fold 4, which shows a strange low accuracy?

Author Response

Please see our point-by-point response in the attached file.

Reviewer 2 Report

In this manuscript, the authors investigated two sequential models from scratch and three pre-trained models, VGG-16, ResNet, and Inception-V3. These models were trained using confocal images of nanoparticle-treated cells loaded with a fluorescent anticancer agent. As a result, several CNN models have been shown to perform substantially well predicting nanoparticle uptake and subsequent drug release into the TNBC cell. Of the pre-trained models, Inception-V3 produced a better outcome with an overall cross-validation accuracy of 99.34%, and hence regarded as a potential model that can be used in future studies in assessing elements of the drug discovery process. Overall, I recommend inviting a major revision of the manuscript before acceptance.

The manuscript should be revised by addressing the issues discussed below: 

  1. Usually, the recognition models of various diseases are presented to solve the problem that the feature difference between some diseases is not easily identified. Accordingly, such models are normally presented many predicted labels (more than five labels) in the confusion matrix. However, in this study, the CNN models only aimed at two predicted labels (either high or low drug uptake and release into TNBC cells). The difference between high and low classes is easily recognized using manual intensity evaluation. Thus, the authors need to claim the necessities and advantages of image recognition using machine learning, compared with the manual intensity evaluation (not only the prediction accuracy comparison). Otherwise, the more predictable labels (not just high and low) must be determined in the model recognition.
  2. In the discussion, the authors claim that VGG-16 performs poorly compared to the other two pre-trained models. However, in the cross-validation accuracy across all models, the performance of ResNet is worse than the VGG-16. Please double-check the presentation of the model performance in the discussion.
  3. Although the pre-trained models were introduced for VGG-16, ResNet-50, and Inception-V3 models, the uneven data caused by a small number of images in a category during training can still lead to poor model predictions. The augmentation type used in this study was rotation. Each applicable patched image was rotated at 45°, resulting in 7 new images. However, there are other ways of data enhancement processing, such as horizontal flips, random rotation, contrast enhancement, brightness enhancement, color enhancement, etc. I'm curious if the prediction performance of the VGG-16 and ResNet models will improve if the augmentation types mentioned above are added.
  4. Please double-check the image numbers throughout the manuscript to avoid confusion in reading the article.

Author Response

Please see our detailed response in the attached file.

Round 2

Reviewer 1 Report

The authors explained more about the novelty and revised some errors. Although the data is still not comprehensive and convincing enough, it seems hard for them to improve in a short time. One major concern is that they only export a result of high or low cell uptake, which highly rely on the setting of threshold. As a researcher in the field of nanomedicine, I won't need AI to tell me it's high or low, but I'd like to obtain a quantitative result.  

Author Response

The authors explained more about the novelty and revised some errors. Although the data is still not comprehensive and convincing enough, it seems hard for them to improve in a short time. One major concern is that they only export a result of high or low cell uptake, which highly rely on the setting of threshold. As a researcher in the field of nanomedicine, I won't need AI to tell me it's high or low, but I'd like to obtain a quantitative result.   

Authors’ Response: While we agree with the reviewer that the quantitative measurement of the signal intensity can be of interest in particular studies for certain purposes, this was outside the scope of the current study and was not the aim of this research. The development of the current models allows us to detect the lead drug candidate based on their cellular uptake for further in-depth analysis among a library of nanotechnology-based or any other platforms. This is a very important information in the early-stage drug discovery field and can be accomplished using our AI models.  

While quantifying the signal intensity was not within the scope of this study, it is possible to achieve this aim if needed in future studies using object detection and/or segmentation models to automatically define the cell region and quantify the signal intensity within the localised region. Although we had already mentioned this capability in the discussion of the manuscript (Line 272), to provide more clarity and accommodate the reviewer's comments, we modified this paragraph and a section in the introduction (Line 112) to read: 

Discussion: Manually removing sections of unimportant details is time-consuming, and there is room for potential bias. A viable approach to this issue is a multi-class classification DL model that can identify for instance the blue signal (cell nucleus), disregard the irrelevant areas, whilst leaving the green signal (payload) as an essential feature for detection. With this approach, pre-processing will not be required before the model can predict unseen images, further removing human bias and making it an efficient method to evaluate cellular drug uptake. Furthermore, other approaches to this hypothesis is to use a model, such as YOLO or other segmentation models (Mask-RCNN, UNet are some), that has localised interested region, ie blue region. The trained models can detect the cancer cell area automatically instead of manually determine areas of interest within the cell where the uptake can be detected and quantified automatically. 

Introduction: Our study aimed to develop a method using DL models that could rationally allow the selection of the best drug candidate based on their cellular uptake and via qualitative analysis, a critical step during the drug development process. 

Round 3

Reviewer 1 Report

Although some flaws are hard to solve in this manuscript, the topic is new and may interest some readers.